# A comparative study on behavior, awareness and belief about cervical cancer among rural and urban women in Vietnam

Minh Tung Phung[1]*, Pham Le An[2,3], Nguyen Nhu Vinh[3], Hong H. T. C. Le[2,4], Karen McLean[5,6], Rafael Meza[1,7], Bhramar Mukherjee[1,8], Alice W. Lee[9], Celeste Leigh Pearce[1]

1 Department of Epidemiology, University of Michigan School of Public Health, Ann Arbor, Michigan, United States of America, 2 Grant and Innovation Center, University of Medicine and Pharmacy at Ho Chi Minh City, Ho Chi Minh City, Vietnam, 3 Family Medicine Training Center, University of Medicine and Pharmacy at Ho Chi Minh City, Ho Chi Minh City, Vietnam, 4 Faculty of Medicine, The University of Queensland, Herston, Queensland, Australia, 5 Division of Gynecologic Oncology, Department of Obstetrics and Gynecology, University of Michigan Medical School, Ann Arbor, Michigan, United States of America, 6 Department of Gynecologic Oncology and Department of Pharmacology & Therapeutics, Roswell Park Comprehensive Cancer Center, Buffalo, New York, United States of America, 7 Department of Integrative Oncology, BC Cancer Research Institute, Vancouver, British Columbia, Canada, 8 Department of Biostatistics, University of Michigan School of Public Health, Ann Arbor, Michigan, United States of America, 9 Department of Public Health, California State University, Fullerton, Fullerton, California, United States of America

☯ These authors contributed equally to this work.
* phungmt@umich.edu

**Data Availability Statement:** There are ethical restrictions which prevent public sharing of minimal data for this study. Restrictions are imposed by the Ethics Committee in Biomedical

## Abstract

Cervical cancer is the second most common gynecologic cancer in Vietnam but based on the literature, only ~25% of Vietnamese women reported ever being screened for cervical cancer. To inform strategies to reduce the cervical cancer burden in Southern Vietnam where disease incidence is higher than the national average, this study examined behaviors, awareness, barriers, and beliefs about cervical cancer screening among rural and urban women in this geographical region. In October-November 2021, we conducted a cross-sectional study among 196 rural and 202 urban women in Southern Vietnam; participants completed a cervical cancer screening questionnaire. Descriptive analyses and rural-urban differences in screening behavior, awareness, barriers, and beliefs are presented. About half of the rural and urban participants reported ever being screened for cervical cancer. Most participants showed high perceived severity of cervical cancer and benefits of screening. Further, they reported that they would screen if it was recommended by doctors and/or friends/family. However, most women showed low awareness and perceived susceptibility to cervical cancer. Logistical and psychosocial barriers to physician-based screening methods were reported. Based on our results, the World Health Organization 2030 goals for cervical cancer screening are not currently met in Southern Vietnam. Increasing health literacy and engaging doctors and family members/social networks emerged as important avenues to improve screening. HPV (*Human papillomavirus*) self-sampling is also a potential approach to increase uptake of cervical cancer screening given the identified psychosocial and logistical barriers.

Research at the Ho Chi Minh City University of Medicine and Pharmacy, Ho Chi Minh City, Vietnam. Data are available from Associate Professor Do Van Dung, PhD, Chair of the Ethics Committee, via email address (dvdung@ump.edu.vn), for researchers who meet the criteria for access to confidential data.

**Funding:** This study was funded by multiple sources from the University of Michigan, including the Rackham International Research Award (to MTP); the Rackham Graduate Student Research Grant (to MTP); the Simson Family Graduate Student Fellowship from the Center for Education for Women (CEW+) (to MTP); the Mary Sue & Kenneth Coleman Student Global Experience Scholarship (to MTP); the Global Public Health Grant for pre-dissertation research from the Office of Global Public Health (to MTP); and the Department of Epidemiology, School of Public Health (to MTP). The funders had no role in study design, data collection and analysis, decision to publish, or preparation of the manuscript.

**Competing interests:** The authors have read the journal's policy and have declared the following competing interests: NNV is a part-time employee for GlaxoSmithKline. Other authors have declared that no competing interests exist. This does not alter our adherence to PLOS ONE policies on sharing data and materials. There are no patents, products in development or marketed products associated with this research to declare.

## Introduction

Cervical cancer is the most common gynecologic cancer among women worldwide with more than 600,000 new cases and 340,000 deaths in 2020 [1]. Approximately 80% of new cases and deaths occur in low- and middle-income countries [2]. Vietnam is a middle-income country [3] where cervical cancer is the second most common gynecologic cancer, with more than 4,000 new cases in 2020 [1]. Cervical cancer risk factors, such as premarital sex and early age at first sexual intercourse, have been on the rise in Vietnam [5, 32], resulting in the disease being considered as a public health priority.

The incidence rate of cervical cancer in urban areas in Southern Vietnam is 1.5–4 times higher than that in Northern urban areas [4, 5]. This is likely a consequence of Vietnam being separated into two nations during the Vietnam War from 1954–1975, during which time Southern Vietnamese people were more exposed to Western culture. Broadly, Western culture is characterized by higher prevalence of cervical cancer risk factors, such as greater number of sex partners and sexually transmitted diseases [5]. Although North and South Vietnam have been reunited for almost 50 years, these sociocultural differences during the war may have had long-lasting effects.

As high-risk *Human papillomavirus* (HPV) is the primary cause of cervical cancer, HPV vaccination has been recommended as the primary prevention strategy for the disease. However, only 8%-12% of Southern Vietnamese women have been vaccinated [6, 7], which is much lower than the World Health Organization (WHO)'s 2030's goal of 90% of girls being fully HPV vaccinated by the age of 15 [8]. Therefore, secondary prevention is important to control the disease.

The long latency period of cervical cancer provides abundant opportunities for early detection [8]. Current available cervical cancer screening methods in Vietnam include cytology (or Papanicolaou "Pap" test), visual inspection with acetic acid (VIA), and HPV testing on physician-collected samples; HPV self-sampling has only recently become available. In the most recent data from a national survey in 2015, ~25% of Vietnamese women aged 18–69 reported ever being screened for cervical cancer [9]. Screening uptake was not reported separately for Southern Vietnam in this survey. Nevertheless, 25% is much lower than the WHO's target for 2030 that 70% of women globally are screened by the age of 35 and again by the age of 45 [8].

Potential factors associated with cervical cancer screening uptake can be grouped into three categories: (1) awareness of the disease and screening, (2) structural barriers such as cost and travel distance, and (3) psychosocial beliefs about the disease and screening [10, 11]. Cancer screening behavior is influenced by individuals' beliefs about their susceptibility of developing cancer, the severity of the disease, the benefits and barriers to screening, and the factors that would make them want to get screened [11, 12]. While some previous studies have assessed the awareness of cervical cancer in Southern Vietnamese people and found a lack of knowledge [6, 13–17], no study has evaluated the barriers and beliefs about cervical cancer and screening in this population. Particularly, rural-urban differences have not been assessed.

To address these gaps in knowledge and to inform strategies to reduce the cervical cancer burden in Southern Vietnam where disease incidence is higher than the national average, we conducted a cross-sectional study to comprehensively assess factors related to cervical cancer screening including awareness, reasons for not screening, and perception toward cervical cancer screening. These analyses were done among rural and urban women separately since socio-economic status, healthcare access and health outcomes in rural areas are poorer than urban areas [18, 19] and hence these two groups may have different screening behaviors, barriers and awareness.

## Materials and methods

### Study design

This study got Institutional Review Board approvals from the University of Michigan (HUM00199150) and the Ho Chi Minh City University of Medicine and Pharmacy (Decisions 446/HDDD-DHYD and 480/HDDD-DHYD). In October and November 2021, we conducted a cross-sectional study of women residing in the rural district of Can Gio and the urban District 4 (both located in the region defined as Ho Chi Minh City, Southern Vietnam). These two districts were selected because they were classified by the local government as being at low risk of COVID-19 during the time of the study. We included women of Kinh ethnicity (accounting for more than 85% of the Vietnamese population [18]) who were aged 30–65 (ages recommended by the WHO to be screened for cervical cancer [8]) with no personal history of cervical cancer.

We worked with the district health center of the rural district of Can Gio to recruit ~200 rural women, including 30% aged 30–39, 30% aged 40–49, and 40% aged 50–65. Can Gio's district health center assigned two community health centers in Can Thanh and Long Hoa to invite 80 and 120 women in their communities, respectively. There are five neighborhoods in Can Thanh; population collaborators (i.e., neighborhood volunteers) from each neighborhood sent invitation letters to 16 women to achieve a total of 80 invited women. There are four neighborhoods in Long Hoa; population collaborators from each neighborhood sent invitation letters to 30 women to achieve a total of 120 invited women. Similarly, we worked with the district health center of the urban District 4 to recruit ~200 urban women. The district health center assigned the two community health centers of Ward 14 and Ward 15 to send invitation letters to 100 women in each of their communities. This number was then divided equally between three neighborhoods in Ward 14 (~33 women each) and four neighborhoods in Ward 15 (25 women each).

Recruitment took place during the weekends and Mondays to maximize the participation opportunity for women who worked outside the home during the weekdays and for those who were busy during the weekends (e.g., housewives), respectively. Some women who were sent invitation letters did not come; information on these women was unavailable to us since we did not have the lists of invitees from the population collaborators. Some women came to our study recruitment sites without an invitation letter. We enrolled all who came to the recruitment sites as long as they met the eligibility criteria.

Female research staff obtained written informed consent from eligible participants. All participants completed a questionnaire in Vietnamese using an electronic tablet, which started with questions on sociodemographic and awareness of cervical cancer. Participants then watched short videos that described the cervical cancer screening methods available in Vietnam: Pap test, VIA, and HPV testing; all videos included voiceover and subtitles in Vietnamese. After watching the videos, women answered questions on their personal experience with each screening method. We asked participants to watch videos to avoid confusions of the screening methods and minimize misclassification. In total, the survey consisted of 112 questions, but there were skip patterns depending on participants' answers. Overall, participation in the study took approximately 30 minutes, and each participant received an incentive of 150,000 Vietnam Dong, which is about half of the average daily individual income in Ho Chi Minh City [20].

### Measures

**Sociodemographic factors.** These factors included self-reported age, religion, education, monthly household income, marital status, having a friend or family member with cervical cancer, having national health insurance, and having private health insurance (Table 1).

**Table 1. Participants' characteristics in rural and urban areas in Southern Vietnam.**

| | Rural | Urban | p-het[a] |
|---|---|---|---|
| | (n = 196) | (n = 202) | |
| **Age** | | | |
| Mean [standard deviation] | 47.4 [9.50] | 47.5 [9.72] | |
| Median [Min, Max] | 47.0 [30.0, 65.0] | 46.0 [30.0, 65.0] | 0.97 |
| **Religion** | | | |
| None | 74 (37.8%) | 45 (22.3%) | |
| Buddhist | 103 (52.6%) | 118 (58.4%) | |
| Other (Christian, Catholic, Caodaiist, Muslim) | 19 (9.7%) | 39 (19.3%) | <0.001 |
| **Education level** | | | |
| Primary school or lower | 75 (38.3%) | 35 (17.3%) | |
| Secondary school | 76 (38.8%) | 78 (38.6%) | |
| High school or higher | 45 (23.0%) | 89 (44.1%) | <0.001 |
| **Monthly household income (million Vietnam Dong)** | | | |
| <5 | 68 (34.7%) | 25 (12.4%) | |
| 5–9.99 | 73 (37.2%) | 69 (34.2%) | |
| 10+ | 55 (28.1%) | 106 (52.5%) | |
| Don't know | 0 (0%) | 2 (1.0%) | <0.001 |
| **Marital status** | | | |
| Never married | 2 (1.0%) | 23 (11.4%) | |
| Married | 159 (81.1%) | 144 (71.3%) | |
| Separated/ Divorced/ Widowed | 35 (17.9%) | 35 (17.3%) | <0.001 |
| **Have friends or family members with cervical cancer** | | | |
| No | 170 (86.7%) | 153 (75.7%) | |
| Yes | 26 (13.3%) | 49 (24.3%) | 0.007 |
| **Have national health insurance** | | | |
| No | 30 (15.3%) | 25 (12.4%) | |
| Yes | 166 (84.7%) | 177 (87.6%) | 0.48 |
| **Have private health insurance** | | | |
| No | 179 (91.3%) | 181 (89.6%) | |
| Yes | 15 (7.7%) | 19 (9.4%) | |
| Don't know | 2 (1.0%) | 2 (1.0%) | 0.82 |

[a] p-heterogeneity between rural and urban areas which are p-values from two sample t-test for continuous variables and chi-squared test for binary/categorical variables.

**Cervical cancer screening uptake.** Participants reported if they had ever been screened for cervical cancer overall and using each screening method (i.e., Pap test, VIA, and HPV testing) and if they had screened within the recommended timeframe. The Vietnam Ministry of Health recommends women aged 21–65 who have had sex to screen every 2 years with Pap test or VIA or every 3 years with HPV testing [21]. Cervical cancer screening uptake factors were coded as binary variables (yes/no; Table 2). Reasons for not screening (i.e., no need/no reasons to screen; not aware of the test; unaffordability; concerns of pain/unpleasant/embarrassment; travelling far; already having been screened using a different method) were coded as binary variables (yes/no; Table 3).

**Awareness of cervical cancer and screening.** We measured if participants correctly identified HPV as the primary cause of cervical cancer (yes/no) and correctly identified the recommended population for cervical cancer screening (yes/no). The recommended population for

**Table 2. Reported cervical cancer screening history by women in rural and urban areas in Southern Vietnam.**

|  | Rural (n = 196)[a] | Urban (n = 202)[a] | p-het[b] |
|---|---|---|---|
| Ever screen | 85 (49.1%) | 102 (51.8%) | 0.69 |
| Pap | 69 (39.9%) | 91 (45.5%) | 0.32 |
| VIA | 15 (8.4%) | 17 (8.7%) | 1.00 |
| HPV testing | 20 (11.4%) | 25 (13.0%) | 0.76 |
| Screen within the recommended timeframe[c] | 60 (35.5%) | 66 (34.0%) | 0.85 |
| Pap | 51 (29.7%) | 55 (27.5%) | 0.73 |
| VIA | 8 (4.6%) | 10 (5.1%) | 1.00 |
| HPV testing | 16 (9.2%) | 18 (9.3%) | 1.00 |

Abbreviations: HPV, *Human papillomavirus*; VIA, visual inspection with acetic acid

[a] Proportion among people with no missing data.

[b] p-heterogeneity between rural and urban areas from chi-squared tests.

[c] Screening timeframe recommended by Vietnam Ministry of Health: Pap or VIA in the past 2 years or HPV testing in the past 3 years.

cervical cancer screening is inconsistent across guidelines. For example, the Vietnam Ministry of Health 2019's guideline recommends women aged 21–65 who have had sex to get screened [21], the U.S. Preventive Services Task Force (USPSTF) 2018's guideline recommends women aged 21–65 regardless of sexual history [22], while the WHO 2021's guideline recommends women aged 30–65 regardless of sexual history [8]. Therefore, answers that matched any of these guidelines were considered correct.

**Beliefs about cervical cancer and screening.** We assessed perceived susceptibility (or one's belief about her risk of getting cervical cancer), perceived severity (or one's belief about the seriousness if she got the disease), perceived benefits (or one's belief about the effectiveness of screening in reducing the threat of the disease), perceived barriers (or one's belief about her obstacles to screening), and cues to action (or one's stimuli that would make her want to get screened).

We adapted questions on perceived susceptibility, severity, benefits, and barriers to cervical cancer screening from the Champion's tool of measuring the Health Belief Model constructs on breast cancer screening, which has been shown to have good validity and reliability [23]. These items are Likert scaled questions with five response options, including: strongly disagree, disagree, neutral, agree and strongly agree. We then grouped these options into three-category variables: disagree (including strongly disagree), neutral, and agree (including

**Table 3. Reporting various reasons for not screening for cervical cancer by women in rural and urban areas in Southern Vietnam.**

|  | Rural (n = 196)[a] | Urban (n = 202)[a] | p-het[b] |
|---|---|---|---|
| No reasons/no need to screen | 85 (50.0%) | 102 (52.8%) | 0.66 |
| Do not know about the screening test | 79 (45.7%) | 84 (44.2%) | 0.86 |
| Too painful, unpleasant, embarrassing | 17 (10.1%) | 18 (9.47%) | 0.99 |
| Too expensive/no insurance/cost | 22 (12.9%) | 15 (7.85%) | 0.16 |
| Have to travel far to screen | 11 (6.6%) | 3 (1.6%) | 0.030 |

[a] Proportion among people without missing values.

[b] p-heterogeneity between rural and urban areas from chi-squared tests.

strongly agree). To access cues to actions, participants were asked if recommendations from doctors and friends/family members as well as being able to screen for free would make them want to get screened for cervical cancer. Each cue to action was coded as a binary variable (yes/no). The items to assess perceived susceptibility, severity, benefits, barriers and cues to action are presented in Table 4.

### Statistical analysis

To identify sociodemographic factors associated with screening uptake (i.e., reporting ever being screened and being screened within the recommended timeframe), we fit logistic regression models separately for the rural and urban areas. These models were regressed on age, religion, education, monthly household income, marital status, having a friend or family member with cervical cancer, having national health insurance and having private health insurance. There were no missing values in these variables, except for monthly household income (proportion of missingness = 0.5%) and having private health insurance (proportion of missingness = 1.0%). Individuals with missing data were not included in the regression models.

Rural-urban differences were assessed in two ways. First, to examine if there were rural-urban differences in sociodemographic factors, screening uptake, reasons for not screening, awareness and beliefs, two sample t-tests (for continuous variables) and chi-squared tests (for categorical and binary variables) were conducted. P-values for heterogeneity (p-het) are provided. Second, to identify rural-urban heterogeneity in the association between the sociodemographic factors and screening uptake (see above), we carried out likelihood ratio tests comparing a model without and a model with an interaction term between each sociodemographic factor and geographic area (rural versus urban).

Statistical significance was defined as $p \leq 0.05$ using two-sided tests. Data were analyzed using R version 4.0.3.

## Results

A total of 196 rural women and 202 urban women enrolled in the study. The mean age of participants in the rural and urban areas was similar (47.4 years in rural; 47.5 years in urban; p-het = 0.97; Table 1). Compared to rural participants, urban participants had higher education (p-het<0.001) and higher household income levels (p-het<0.001; Table 1). Urban participants were also more likely to have friends or family members with cervical cancer (p-het = 0.007), more likely to be religious (p-het<0.001), but less likely to be married (p-het<0.001; Table 1).

### Cervical cancer screening uptake and reasons for not screening

About half of the participants reported ever being screened for cervical cancer (49.1% in rural; 51.8% in urban), and a third reported being screened within the recommended timeframe (35.5% in rural; 34.0% in urban; Table 2). Among people 30–35 years old, 50% of rural and 32% of urban participants reported ever being screened, however the sample size in the age group was small (n = 22 and n = 25 in rural and urban areas, respectively). Pap test was the most common screening method reported, with 39.9% of rural women and 45.5% of urban women reporting ever being screened using this method. None of these comparisons was statistically significantly different between the rural and urban groups (Table 2).

Overall, both rural and urban participants with either private or national health insurance were more likely to report ever being screening (having private health insurance odds ratio OR = 2.65, 95% confidence interval CI 1.02–6.87; having national health insurance OR = 4.15, 95% CI 1.93–8.93) and being screening within the recommended time frame (having private health insurance OR = 2.33, 95% CI 0.94–5.76; having national health insurance OR = 4.46,

**Table 4.  Beliefs about cervical cancer and screening among women in rural and urban areas in Southern Vietnam.**

| | Rural | Urban | p-het[b] |
|---|---|---|---|
| | (n = 196)[a] | (n = 202)[a] | |
| **Perceived susceptibility to cervical cancer** | | | |
| I feel I will get cervical cancer in the future. | | | |
| Disagree | 121 (61.7%) | 117 (57.9%) | |
| Neutral | 32 (16.3%) | 38 (18.8%) | |
| Agree | 43 (21.9%) | 47 (23.3%) | 0.72 |
| I am more likely than the average women to get cervical cancer. | | | |
| Disagree | 153 (78.9%) | 133 (65.8%) | |
| Neutral | 25 (12.9%) | 36 (17.8%) | |
| Agree | 16 (8.3%) | 33 (16.3%) | 0.005 |
| **Perceived severity of cervical cancer** | | | |
| When I think about cervical cancer, my heart beats faster. | | | |
| Disagree | 68 (34.9%) | 83 (41.1%) | |
| Neutral | 22 (11.3%) | 13 (6.44%) | |
| Agree | 105 (53.8%) | 106 (52.5%) | 0.16 |
| I am afraid to think about cervical cancer. | | | |
| Disagree | 44 (22.6%) | 44 (21.8%) | |
| Neutral | 5 (2.56%) | 7 (3.47%) | |
| Agree | 146 (74.9%) | 151 (74.8%) | 0.86 |
| Problems I would experience with cervical cancer would last a long time. | | | |
| Disagree | 30 (15.4%) | 18 (8.91%) | |
| Neutral | 14 (7.18%) | 18 (8.91%) | |
| Agree | 151 (77.4%) | 166 (82.2%) | 0.13 |
| Cervical cancer would threaten the relationship with my boyfriend, husband or partner. | | | |
| Disagree | 45 (23.1%) | 34 (16.8%) | |
| Neutral | 14 (7.18%) | 18 (8.91%) | |
| Agree | 136 (69.7%) | 150 (74.3%) | 0.27 |
| If I had cervical cancer my whole life would change. | | | |
| Disagree | 33 (17.0%) | 25 (12.4%) | |
| Neutral | 13 (6.70%) | 13 (6.44%) | |
| Agree | 148 (76.3%) | 164 (81.2%) | 0.41 |
| If I developed cervical cancer, I would not live longer than 5 years. | | | |
| Disagree | 63 (32.3%) | 78 (38.6%) | |
| Neutral | 30 (15.4%) | 34 (16.8%) | |
| Agree | 102 (52.3%) | 90 (44.6%) | 0.29 |
| **Perceived benefits of cervical cancer screening** | | | |
| When I undergo cervical cancer screening I feel good about myself. | | | |
| Disagree | 2 (1.0%) | 1 (0.5%) | |
| Neutral | 1 (0.5%) | 1 (0.5%) | |
| Agree | 193 (98.5%) | 200 (99.0%) | 0.83 |
| When I complete cervical cancer screening I don't worry as much about cervical cancer. | | | |
| Disagree | 12 (6.2%) | 7 (3.5%) | |
| Neutral | 2 (1.0%) | 3 (1.5%) | |
| Agree | 181 (92.8%) | 192 (95.0%) | 0.42 |
| Completing cervical cancer screening will allow me to find out if there are early signs of cervical cancer. | | | |
| Disagree | 6 (3.1%) | 2 (1.0%) | |
| Neutral | 7 (3.6%) | 6 (3.0%) | |

*(Continued)*

**Table 4.** (Continued)

| | Rural | Urban | p-het[b] |
|---|---|---|---|
| | (n = 196)[a] | (n = 202)[a] | |
| Agree | 183 (93.4%) | 194 (96.0%) | 0.32 |
| If I complete cervical cancer screening, I will decrease my chances of dying from cervical cancer. | | | |
| Disagree | 12 (6.2%) | 8 (4.0%) | |
| Neutral | 1 (0.5%) | 13 (6.4%) | |
| Agree | 182 (93.3%) | 181 (89.6%) | 0.004 |
| If I complete cervical cancer screening, I will decrease my chance of pain and surgery related to cervical cancer. | | | |
| Disagree | 7 (3.59%) | 8 (3.96%) | |
| Neutral | 5 (2.56%) | 10 (4.95%) | |
| Agree | 183 (93.8%) | 184 (91.1%) | 0.45 |
| If I complete cervical cancer screening, it will help me find abnormality in my cervix which can become cancer. | | | |
| Disagree | 4 (2.04%) | 2 (0.990%) | |
| Neutral | 1 (0.510%) | 5 (2.48%) | |
| Agree | 191 (97.4%) | 195 (96.5%) | 0.19 |
| **Perceived barriers to cervical cancer screening** | | | |
| I feel awkward going to get cervical cancer screening. | | | |
| Disagree | 146 (74.5%) | 164 (81.2%) | |
| Neutral | 2 (1.02%) | 9 (4.46%) | |
| Agree | 48 (24.5%) | 29 (14.4%) | 0.006 |
| Undergoing cervical cancer screening will make me worry about cervical cancer. | | | |
| Disagree | 94 (48.0%) | 80 (39.6%) | |
| Neutral | 6 (3.06%) | 10 (4.95%) | |
| Agree | 96 (49.0%) | 112 (55.4%) | 0.20 |
| Cervical cancer screening will be embarrassing to me. | | | |
| Disagree | 166 (84.7%) | 180 (89.1%) | |
| Neutral | 0 (0%) | 6 (3.0%) | |
| Agree | 30 (15.3%) | 16 (7.9%) | 0.005 |
| Undergoing cervical cancer screening will take too much time. | | | |
| Disagree | 137 (69.9%) | 144 (71.3%) | |
| Neutral | 4 (2.04%) | 12 (5.9%) | |
| Agree | 55 (28.1%) | 46 (22.8%) | 0.087 |
| Cervical cancer screening procedure will be painful. | | | |
| Disagree | 76 (38.8%) | 84 (41.6%) | |
| Neutral | 19 (9.69%) | 20 (9.9%) | |
| Agree | 101 (51.5%) | 98 (48.5%) | 0.83 |
| Cervical cancer screening will interfere with my family obligations | | | |
| Disagree | 138 (70.4%) | 163 (80.7%) | |
| Neutral | 2 (1.02%) | 7 (3.5%) | |
| Agree | 56 (28.6%) | 32 (15.8%) | 0.003 |
| **What would make you want to get screened for cervical cancer?** | | | |
| Your doctor recommending screening. | | | |
| No | 41 (20.9%) | 18 (8.91%) | |
| Yes | 155 (79.1%) | 184 (91.1%) | 0.001 |
| Being able to do it for free. | | | |
| No | 48 (24.5%) | 40 (19.8%) | |
| Yes | 148 (75.5%) | 162 (80.2%) | 0.31 |
| A friend or family member telling you why screening is important. | | | |

*(Continued)*

**Table 4.** (Continued)

| | Rural | Urban | p-het[b] |
|---|---|---|---|
| | (n = 196)[a] | (n = 202)[a] | |
| No | 66 (33.7%) | 47 (23.3%) | |
| Yes | 130 (66.3%) | 155 (76.7%) | 0.028 |

[a] Numbers may not sum to total due to missing values.

[b] p-heterogeneity between rural and urban areas from chi-squared tests.

95% CI 1.77–11.25) compared to people without those insurance types. Also, participants with at least a high school education were more likely to report having been ever screened. No other sociodemographic factors were associated with screening history.

In both rural and urban areas, "no reasons/no need to screen" was the most common reason for not being screened for cervical cancer (50.0% in rural; 52.8% in urban; p-het = 0.66; Table 3). The second most common reason was "do not know about the screening test" (45.7% in rural; 44.2% in urban; p-het = 0.86). Pain and embarrassment were reported by 10% of participants as the reason for not screening whereas cost was the reason for 13% of rural and 8% of urban participants. Rural women were more likely to indicate that having to travel far was a reason for not screening compared to urban women (6.6% in rural; 1.6% in urban; p-het = 0.030; Table 3).

A third of participants correctly identified that HPV infection was the primary cause of cervical cancer (28.1% in rural; 34.7% in urban; p-het = 0.19). Compared to rural women, urban women were more likely to correctly know the population recommended for cervical cancer screening (46.4% in rural; 66.8% in urban; p-het<0.001).

## Beliefs about cervical cancer and screening

Less than a quarter of participants felt that they would get cervical cancer in the future (21.9% in rural; 23.3% in urban; p-het = 0.72; Table 4). A smaller proportion of participants believed they were more likely to develop cervical cancer compared to the average woman; however, urban women were more likely to believe this compared to rural women (8.3% in rural; 16.3% in urban; p-het = 0.005; Table 4).

Overall, most women believed that cervical cancer would cause serious problems in their lives if they were diagnosed with it. At least 70% of participants were afraid to think about cervical cancer, thought that problems with cervical cancer would last for a long time and that the disease would threaten their relationships with their partners and would change their whole life (p-het>0.05 for rural-urban differences; Table 4).

The majority of women felt that screening was effective in reducing the harmful effects of cervical cancer. At least 90% of women believed that screening would make them feel good about themselves. A similar proportion of participants felt that screening would allow them to find early signs of cervical cancer as well as abnormality in their cervix, and would decrease their chance of pain, surgery and mortality related to the disease (Table 4).

The most common perceived barriers to screening were that screening would make them worry about cervical cancer (49.0% in rural; 55.4% in urban; p-het = 0.20) and that the screening procedure would be painful (51.5% in rural; 48.5% in urban; p-het = 0.83; Table 4). Compared to urban women, rural women were more likely to feel that being screened for cervical cancer was awkward (24.5% in rural; 14.4% in urban; p-het = 0.006), embarrassing (15.3% in rural; 7.9% in urban; p-het = 0.005), and would interfere with their family obligations (28.6% in rural; 15.8% in urban; p-het = 0.003; Table 4).

Women in both areas indicated that their doctors' recommendation was the most important factor that would make them want to get screened for cervical cancer, followed by being able to screen for free, and then recommendations from friends/family members. Compared to rural women, urban women were more likely to indicate that they would want to get screened if receiving recommendations from doctors (79.1% in rural; 91.1% in urban; p-het = 0.001) or friends/family members (66.3% in rural; 76.7% in urban; p-het = 0.028; Table 4).

## Discussion

In this study about cervical cancer screening in Southern Vietnam, we found that ~50% of women reported ever being screened for cervical cancer, which is higher than the finding from the 2015 national survey that indicated ~25% of Vietnamese women aged 18–69 reported having ever screened [24]. It is possible that participants in the national survey did not know what cervical cancer screening was and therefore did not correctly report their history. We asked our participants to watch videos showing screening methods to minimize misclassification. It is also possible that the areas in which we conducted our study have higher screening rates than other areas in Vietnam. Also, there may have been a temporal shift in screening prevalence since 2015. Regardless, the screening percentages are much lower than the WHO's 2030 target that 70% of women in each country are screened by the age of 35 as we observed only 40% of women in the 30–35 age group had ever screened.

Our study identified three important beliefs that would facilitate the increased uptake of cervical cancer screening in Southern Vietnam. First, women showed a strong belief that cervical cancer would cause serious problems in their lives if they were diagnosed with it and that screening was effective in reducing the harmful effects of cervical cancer. Second, participants indicated that a doctor's recommendation to screen would be impactful; about 80% of participants indicated that they would get screened if recommended by their doctors. Therefore, it is important to engage physicians in interventions to improve cervical cancer screening. Lastly, ~70% of participants in our study indicated that they would get screened if recommended by friends or family members, which suggests that future interventions should engage women's social networks to improve screening uptake.

However, we identified several challenges to cervical cancer screening. First, we found that women had poor awareness of cervical cancer, which is consistent with previous studies in Southern Vietnam [6, 13–17]. In our study, only about a half of women knew the recommended screening population, which is in line with our other finding that the most common reason for not screening is "not reasons/no need to screen". Second, the majority of women felt that they were not at risk of cervical cancer. This is a challenge to screening because women who believe they are at risk of cervical cancer are more likely to screen [11]. The third challenge is that participants indicated several logistical and psychosocial barriers to cervical cancer screening, including screening making them worry about cancer as well as screening being painful, awkward, embarrassing, and time consuming, particularly among rural women. Tailored health education programs are needed to reduce these psychosocial barriers. Another potential solution is HPV self-sampling at home since this would obviate visiting healthcare providers thereby addressing the concerns related to embarrassment and travel time to an appointment. HPV self-sampling has been shown to improve cervical cancer screening uptake in urban and rural settings, in low- and middle-income countries, and hard-to-reach populations [25–28]. It may be a viable approach in Southern Vietnam as it has been successfully utilized in Guatemala [29, 30], Thailand [31], and an urban area in Northern Vietnam [32].

Strengths of this study include measures that were undertaken to minimize information bias. We used electronic tablets to deliver the questionnaire instead of an interview to ensure the privacy of participants' answers. In addition, we used videos describing the screening methods to minimize participant confusion. A limitation of this study is that we did not have access to population rosters, and thus we were unable to determine if participants were different from non-participants in these two areas. It is also possible that the lack of rural-urban differences observed in our study was due to the small sample size. Our rural study sample had similar education levels as the general rural population in Ho Chi Minh City, but our urban study sample had lower education levels when compared to the general urban population in Ho Chi Minh City [18]. Therefore, our study's findings for urban women may not be generalizable to urban women with different education levels. Also, due to the COVID-19 pandemic, we were restricted from carrying out this study in a more remote rural setting. Future research should include more rural settings to ensure generalizability to this group.

## Conclusions

Given the increase in cervical cancer risk factors in Vietnam [5, 33] and cervical cancer being the second most common gynecologic cancer in Vietnam, it is a public health priority to reduce the burden of cervical cancer in Vietnamese women. This is particularly the case in Southern Vietnam where disease incidence is higher than the national average. Our study is the first to comprehensively examine cervical cancer screening behaviors, awareness, barriers, and beliefs in rural and urban areas in Southern Vietnam. We found low reported screening in both areas and identified opportunities and challenges to promoting cervical cancer screening which include the importance of improving health literacy as well as engaging doctors and family members/social networks. Novel approaches such as home-based HPV self-sampling may be an alternative strategy to increase uptake of cervical cancer screening given the identified psychosocial and logistical barriers related to physician-based screening.

## Acknowledgments

We would like to thank the leadership and staff at the Departments of Health of Ho Chi Minh City, Can Gio District and District 4, the community health centers at Can Thanh and Long Hoa (Can Gio District) and at Ward 14 and Ward 15 (District 4) for helping us conduct this study. We would also like to thank the research staff, neighborhood volunteers and participants.

## Author Contributions

**Conceptualization:** Minh Tung Phung, Pham Le An, Karen McLean, Rafael Meza, Bhramar Mukherjee, Celeste Leigh Pearce.

**Data curation:** Minh Tung Phung.

**Formal analysis:** Minh Tung Phung.

**Funding acquisition:** Minh Tung Phung, Celeste Leigh Pearce.

**Investigation:** Minh Tung Phung, Pham Le An, Celeste Leigh Pearce.

**Methodology:** Minh Tung Phung, Pham Le An, Celeste Leigh Pearce.

**Project administration:** Minh Tung Phung, Pham Le An, Hong H. T. C. Le, Celeste Leigh Pearce.

**Resources:** Pham Le An, Celeste Leigh Pearce.

**Supervision:** Pham Le An, Celeste Leigh Pearce.

**Writing – original draft:** Minh Tung Phung, Alice W. Lee, Celeste Leigh Pearce.

**Writing – review & editing:** Minh Tung Phung, Pham Le An, Nguyen Nhu Vinh, Hong H. T. C. Le, Karen McLean, Rafael Meza, Bhramar Mukherjee, Alice W. Lee, Celeste Leigh Pearce.

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
