## [Decision Letter · Decision Letter 0]

2 Mar 2023

PGPH-D-23-00181

A comprehensive assessment of cervical cancer screening among women in rural and urban areas in Southern Vietnam

Dear Dr. Phung,

Thank you for submitting your manuscript to PLOS Global Public Health. After careful consideration, we feel that it has merit but does not fully meet PLOS Global Public Health’s publication criteria as it currently stands. Therefore, we invite you to submit a revised version of the manuscript that addresses the points raised during the review process.

We look forward to receiving your revised manuscript.

Kind regards,

Nnodimele Onuigbo Atulomah, PhD

Academic Editor

Journal Requirements:

2. Please send a completed 'Competing Interests' statement, including any COIs declared by your co-authors. If you have no competing interests to declare, please state "The authors have declared that no competing interests exist". Otherwise please declare all competing interests beginning with the statement "I have read the journal's policy and the authors of this manuscript have the following competing interests:"

3. Please amend your detailed Financial Disclosure statement. This is published with the article. It must therefore be completed in full sentences and contain the exact wording you wish to be published.

4. Please ensure that Funding Information and Financial Disclosure Statement are matched.

5. We noticed that you used "data not shown and unpublished data" in the manuscript. We do not allow these references, as the PLOS data access policy requires that all data be either published with the manuscript or made available in a publicly accessible database. Please amend the supplementary material to include the referenced data or remove the references.

Additional Editor Comments:

The reviewers have appraised the manuscript and reported that the theme of the study is very contemporary and relevant to diagnose an important health behaviour; Cervical cancer screening of women in selected communities in southern Vietnam. I totally agree with the three reviewers' observations and recommend that the authors give due attention to their suggestions. I shall comment on their observations but mention areas they have inadvertently omitted. In addition, I have made some observations and recommendations that would strengthen the manuscript to interest readers when published.

TITLE: This title provides limited information about the study and would greatly benefit from modifications considering that the theoretical underpinning of the problem phenomenon concerns in cervical cancer is screening in early diagnosis stage of secondary prevention, which is scientifically offered by the health belief model and public health principles of prevention and control.

Strongly Suggested title: "Health-Belief constructs as predictors of cervical cancer screening among women in selected rural and urban communities of Southern Vietnam" With short title as; "Short title: Cervical cancer screening in rural and urban Southern Vietnam".

Of course, "comprehensive assessment" need not be included in the title because the observational study is already an assessment of the problem phenomenon and seeks to elucidate the underpinning dynamics. Again the application of the theoretical framework has defined the rigor involved.

ABSTRACT: This needs to be appropriately written according to the required format for PLoS Global Public Health. All necessary information should be included Background, Objectives, Methods, Results and Findings with keywords.

INTRODUCTION: All aspects of the introduction appear good showing the epidemiological principles but little of risk factors has been mentioned, public health principles of prevention and control with their respective modes of intervention are completely omitted but very essential to establish the underpinning public health scientific principles. Importantly, lines 81- 91 where the paper has introduced The Health Belief Model as theoretical framework underpinning health behaviour and health-seeking in health-risk condition typified by cancer of the cervix is weak and should be visible in the title because the construct provides the validity of the measurement in the study. The mention of this theoretical framework strengthens the scientific basis of the study, but it is also observed that this was casually mentioned and poorly structured to guide the entire study.

Line 92: The objectives set for the study includes: evaluate level of knowledge about cancer of the cervix among women in urban-rural communities of southern Vietnam, and to inform strategies to reduce the cervical cancer in the region. But knowledge is not sufficient to trigger health-seeking behaviour, other antecedent variables in the theoretical framework are not set as course of action or specific objectives to guide the study. What factors are referred to in line 94 which are not contained in the theoretical framework?

Observed that The theoretical framework was poorly operationalized in the study.

METHODOLOGY: The cross-sectional study did not mention clearly the populations of the communities selected and sample size computation to indicate the appropriate sample size for such survey considering the implications of generalizability of the results obtained as the statements in the study concerning prevalence of the disease in the region.

Line 133-134 may have introduced some bias Participants then watched short videos that described the cervical cancer screening methods available in Vietnam.

Data analysis needs to be adequately organized to mention all variables in the study and how these are measured. For example; cervical cancer screening uptake represents health-seeking at what time and what symptoms trigger desire to go for a test and how are these scored. The tables all have frequency distributions for every items in the instrument. This makes the result boring to read and confusing. The scores of the participants regarding moderating variables in the theoretical framework are absent even thought study claimed to have collected them.

With the Likert scale weighted-aggregate score for each participant would have compressed all item to a single score for each participant and the mean with standard deviation synthesized. Statistical analysis in the study are grossly inadequate.

DISCUSSION: The discussion should not begin with the statement "..the first study to comprehensively examine cervical cancer…". what constitute comprehensive in the study?

Line 265 "we found that ~50% of women reported ever being screened for cervical cancer…" contradict statement in line 36-37 "only a few Vietnamese women reported ever being screened for cervical cancer". About 50% cannot be regarded as few. What makes it comprehensive when variables such as attitudinal dispositions of respondents, perceived benefits, self-efficacy expectations of taking a screening were not considered within the health belief construct.

CONCLUSION: To state in the conclusion in line 316 that lack of a national screening program in the study area contradicts "Screening timeframe recommended by Vietnam Ministry of Health" and other risk factors for the disease is a priority for intervention is inappropriate. No where in the introduction did the author mention that national screening programme is a challenge. Health literacy was never a concern for the study, and why should this be part of the conclusion.

Reviewers' comments:

Reviewer's Responses to Questions

**Comments to the Author**

1. Does this manuscript meet PLOS Global Public Health’s publication criteria? Is the manuscript technically sound, and do the data support the conclusions? The manuscript must describe methodologically and ethically rigorous research with conclusions that are appropriately drawn based on the data presented.

Reviewer #1: Yes

Reviewer #2: Yes

Reviewer #3: Yes

2. Has the statistical analysis been performed appropriately and rigorously?

Reviewer #1: Yes

Reviewer #2: Yes

Reviewer #3: Yes

3. Have the authors made all data underlying the findings in their manuscript fully available (please refer to the Data Availability Statement at the start of the manuscript PDF file)?

Reviewer #1: No

Reviewer #2: No

Reviewer #3: Yes

4. Is the manuscript presented in an intelligible fashion and written in standard English?

Reviewer #1: Yes

Reviewer #2: Yes

Reviewer #3: Yes

5. Review Comments to the Author

Reviewer #1: The article is scientific in nature; also the presentation of tables and discussions of findings demonstrated the nature of an evidence-based study. The study has added to the existing body of knowledge and literature within the scope of this study.

Note that, the authors have requested for a waver on data sharing requirement.

Reviewer #2: The article is very well written and factual. It presents evidence of the first comprehensive evaluation of cervical cancer screening behaviors, beliefs and attitudes in rural and urban areas in Southern Vietnam, where the incidence of cervical cancer is higher than the national average. This is commendable. The authors should: (1) correct a typographical error on page 5 line 68 (Replace sexual with sexually); (2) recompute the summary statistic on women who had undergone cervical cancer screening by the age of 35 years ( see page 15 line 274).

Reviewer #3: Dear Editor-in- Chief,

I have provided my comments on the manuscript, PGPH-D-23-00181 titled, “A comprehensive assessment of cervical cancer screening among women in rural and urban areas in Southern Vietnam. The manuscript presents a comparative assessment of cervical cancer screening among women in rural and urban areas in Ho Chin Minh City Southern Vietnam. The manuscript is generally well written and presented.

Other general and specific observations follow:

1. Originality of value: The manuscript presents an original study that was designed and conducted among rural and urban women aged over 30-65 years in Vietnam. The paper addressed a pertinent public health issue, namely cervical cancer screening with fascinating results.

2. Suitability and soundness of technique: The techniques used for the study are appropriate and robust. It utilized cross-sectional design and participants were recruited through invitation

3. Clarity of Presentation: The manuscript was very clear in narrative presentation. It is easy to follow and complied with the journal specifications.

4. Areas requiring correction: While the paper is generally well written, some corrections are required, which have been highlighted for the authors' attention below.

Title

I am suggesting a change in the title of the article for clear understanding of the study.

The title could be change to “A comparative study on behavior, awareness and belief about cervical cancer among rural and urban women in Vietnam”. This captured the results presented clearly.

Abstract

Line 39-40: This statement should be written as “this study examined behavior, awareness and beliefs about cervical cancer screening among rural and urban women in this geographical region. The cross-sectional study should be removed, since it appeared in the methods.

Line 54: it will be good to suggest approaches to increase the uptake of cervical cancer screening in that locality instead of stating that novel approaches are needed to increase uptake of cervical cancer. Since one of the study focus was to inform strategies to reduce cervical cancer burden in Southern Vietnam.

Material and Methods

The authors did not specify how the sample size was calculated. The type of sampling technique used was not clearly stated.

Line 133: The authors stated that “participants watched short video that described cervical cancer screening before they filled the questionnaire”. I don’t think this is appropriate with the study design. I will like the authors to provide justification for during this.

Conclusion:

Line 319: The authors concluded that they examined cervical cancer screening behavior, belief, and attitude. The results of the study did not cover attitude.

6. PLOS authors have the option to publish the peer review history of their article (what does this mean?). If published, this will include your full peer review and any attached files.

**Do you want your identity to be public for this peer review?** For information about this choice, including consent withdrawal, please see our Privacy Policy.

Reviewer #1: **Yes: **Khadijat Toyin Musah PhD

Reviewer #2: No

Reviewer #3: **Yes: **Titilayo Olaoye

---

## [Editor Report · Decision Letter 1]

17 Apr 2023

A comparative study on behavior, awareness and belief about cervical cancer among rural and urban women in Vietnam

PGPH-D-23-00181R1

Dear Dr. Phung,

We are pleased to inform you that your manuscript 'A comparative study on behavior, awareness and belief about cervical cancer among rural and urban women in Vietnam' has been provisionally accepted for publication in PLOS Global Public Health.

Best regards,

Nnodimele Onuigbo Atulomah, PhD

Academic Editor

Having reviewed the responses of the authors to the issues raised by the reviewers, I confirm that all have been presented to the satisfaction of the Academic Editor assigned to handle the manuscript. On this I hereby recommend that the Editor in chief consider the manuscript for publication.